# The Effect of Frailty on Body Composition and Its Impact on the Use of SGLT-2 Inhibitors and GLP-1RA in Older Persons with Diabetes

**DOI:** 10.3390/metabo15060381

**Published:** 2025-06-09

**Authors:** Alan Sinclair, Izel Siqueira, Ahmed Abdelhafiz

**Affiliations:** 1King’s College, London WC2R 2LS, UK; alan.sinclair@kcl.ac.uk; 2Foundation for Diabetes Research in Older People (fDROP), Droitwich Spa WR9 0QH, UK; 3Department of Geriatric Medicine, Rotherham General Hospital, Moorgate Road, Rotherham S60 2UD, UK; izel.siqueira1@nhs.net

**Keywords:** older persons, frailty, body composition, diabetes mellitus, SGLT-2 inhibitors, GLP-1RA

## Abstract

The association of frailty with body composition is complex. Frailty can be associated with significant anorexia and weight loss or overweight and obesity. In addition, the development of frailty leads to changes in muscle mass, muscle fibre type, and visceral fat. In older people with diabetes, frailty-induced body composition changes are clinically relevant as they may affect the metabolic profile of the frail person. The determinants of insulin resistance in frail older persons with diabetes include factors such as total body weight and the ratio of lean muscle mass to visceral fat mass. The predominant loss of insulin-resistant skeletal muscle fibres type II, in comparison to insulin-sensitive type I fibres, is another factor that modifies the overall insulin resistance of the individual. As a result, frailty appears to be a heterogeneous condition with variable insulin resistance across a metabolic spectrum. The spectrum spans from a sarcopenic obese frail phenotype at one end to an anorexic malnourished frail phenotype at the other end. The introduction of SGLT-2 inhibitors and GLP-1RA with novel anti-metabolic syndrome properties, not just glucose-lowering effect, should influence clinicians’ choice in frail older persons with diabetes. These agents are likely to be beneficial in patients at the sarcopenic obese end of the frailty spectrum, who should benefit most due to their high baseline risk of progression of metabolic syndrome, high insulin resistance, and the increased prevalence of cardiovascular risk factors. On the other hand, patients at the anorexic malnourished end of the frailty spectrum are likely not suitable for such therapy due to the regression of metabolic syndrome in this group of patients and the increased risk of further weight loss, dehydration, and hypotension.

## 1. Introduction

The global prevalence of diabetes is increasing from 8.4% in 2017 to an estimated 10% in 2045 [1]. The increase is particularly prevalent in older age groups due to increased life expectancy, and the peak reaches 24% for those aged 75–79 years [1]. In older persons, frailty is emerging as a new and clinically important complication of diabetes [2]. Several studies have demonstrated the increased prevalence of frailty in diabetes, the association of diabetes with increased risk of frailty, and the high burden of frailty in older persons living with diabetes [3,4,5]. Frailty is usually seen as a wasting or shrinkage disease, which increases the risk of hypoglycaemia [6]. The new therapies of sodium glucose cotransporter-2 (SGLT-2) inhibitors and glucagon-like peptide-1 receptor agonists (GLP-1RA) are associated with significant weight loss. Therefore, the current guidelines recommend relaxed glycaemic targets in frail older people with diabetes and cautious use of SGLT-2 inhibitors and GLP-1RA [7,8]. Frailty is not well defined in clinical guidelines and is usually referred to as one homogeneous group of patients. However, frailty appears to affect the body composition in a way that may alter the metabolic profile of the individual. Therefore, frailty is a metabolically heterogeneous condition and spans a spectrum of populations with variable insulin resistance due to the variable effects of frailty on body composition. In other words, frailty-associated metabolic heterogeneity dictates SGLT-2 inhibitors and GLP-1RA suitability in older persons with diabetes. As a result, the use of SGLT-2 inhibitors and GLP-1RA should not be generalised but guided by the underlying metabolic profile of the individual to help clinicians to practice a precision medicine approach. This manuscript reviews age-related body composition changes and the effects of frailty on body composition and explores the implications of the use of SGLT-2 inhibitors and GLP-1RA in frail older persons with diabetes.

## 2. Body Composition—The Ageing Effect

Ageing is associated with an increase in total body weight by about 0.3–0.5% per year after the age of 40 years [9]. Body composition also changes with increasing age, with a decline in skeletal muscle mass and an increase in total body fat [10]. The decline in muscle mass is progressive at an annual rate of 1–2% after the age of 30 years, accelerates to 1.5–3% per year after the age of 60 years, and continues to accelerate after the age of 75 years [11]. Fat distribution also changes with an increase in visceral fat, ectopic fat deposition in body organs, including skeletal muscles, and a reduction in subcutaneous fat [12]. The decline in skeletal muscle mass, the main site of glucose consumption, and the increase in visceral and ectopic fat may substantially alter insulin resistance and glucose metabolism in old age. A sedentary lifestyle associated with ageing may further increase body weight and visceral fat and reduce muscle mass [13]. An age-related decline in oestrogen and testosterone reduces muscle mass and increases visceral fat [14]. Blood glucose levels are dependent on the rate of glucose entry from intestinal absorption, glycogenolysis or gluconeogenesis, and glucose removal from the circulation by organ consumption. In addition to insulin, blood glucose levels are maintained by other regulatory hormones such as glucagon, gastric inhibitory polypeptide (GIP), glucagon-like peptide-1 (GLP-1), epinephrine, cortisol, and growth hormone [15]. Ageing is associated with a reduction in insulin secretion by 0.7% per year due to a combination of reduced β-cell function and number [16]. Disturbances in the physiology of the gut-derived incretins GIP and GLP-1 may be involved in β-cell dysfunction [17]. These peptides enhance insulin secretion after meals and may have a role in the maintenance of β-cell growth, proliferation, and inhibition of apoptosis. Ageing is associated with reduced levels and function of these peptides [18]. Therefore, ageing is associated with a disturbed muscle–fat balance, which may increase insulin resistance and alter glucose metabolism. However, the increase in body fat, particularly visceral fat, is much more predictor of insulin resistance than just ageing alone [19].

## 3. Body Composition—The Frailty Effect

Frailty is a state of reduced physiologic reserve at multiple organ level that increases vulnerability to physical or psychological stressors [20]. The organ systems involved are neuroendocrine, cardiovascular, immune, metabolic, and nervous systems [21]. Diabetes increases the risk of frailty due to diabetes-related complications and diabetes-associated morbidities, especially hypertension and chronic kidney disease [22,23]. Frailty reciprocally increases the risk of diabetes [24]. The occurrence of chronic low-grade inflammation could be the common factor contributing to the association of frailty and diabetes [25]. The relationship between frailty and body weight and body composition is complex, as explained below.

### 3.1. Frailty and Obesity

While diagnostic criteria of frailty syndrome include weight loss along with reduced activity, slow gait, weakness of hand grip strength, and exhaustion, weight loss is not fundamental for frailty diagnosis [26]. Weight gain has also been described in frail people with significant muscle mass loss consistent with a sarcopenic obese frailty phenotype [27]. Overweight or obesity appears to be associated with higher levels of frailty. In a study to examine the associations between high body mass index (BMI) with frailty in older persons, BMI levels ≥ 25 kg/m^2^ were associated with higher frailty compared to normal BMI (*p* < 0.001). Higher percent body fat was also associated with higher frailty (*p* < 0.001). Percent body fat significantly mediated the relationship between BMI levels and frailty. Therefore, the relationship between BMI and frailty is partially explained by body fat [28]. In a Chinese study, participants with high visceral fat area (VFA) showed a higher risk of prefrailty {adjusted odds ratio (OR) 1.53, 95% confidence interval (CI) 1.19 to 1.96}. However, having an overweight BMI with normal VFA was a protective factor for prefrailty (OR 0.62, 95% CI 0.43 to 0.90), but having a normal weight but excess VFA increased the risk of prefrailty (OR 1.87, 95% CI 1.15 to 3.03), suggesting that visceral fat obesity is an independent risk factor for prefrailty [29]. A previous meta-analysis also showed that high BMI (≥35.0 kg/m^2^) is associated with a higher probability of frailty [30]. This suggests that both general obesity (high BMI) and central obesity (high VFA) are linked to a higher risk of frailty [31]. In another study, prefrailty and frailty were linked to higher levels of adiposity in older adults only when both general obesity (BMI ≥ 30.0 kg/m^2^) and abdominal obesity {waist circumference (WC) > 102 cm for men and >88 cm for women} were present. This emphasises the importance of the evaluation of both BMI and WC in clinical practice. This also suggests that older persons presenting with both general and abdominal obesity should be routinely screened for frailty [32]. Furthermore, it has been shown that both general and abdominal obesity during adulthood are associated with an increased risk of prefrailty/frailty in later years [33].

### 3.2. Frailty and Underweight

Frailty is often viewed as a syndrome characterised by unintentional weight loss and weakness that leads to impaired mobility, balance, and minimal energy reserve [26]. Underweight older persons are well known to be vulnerable, have minimal reserve capacity, and become frail [26]. They also have an increased risk for osteoporosis, impaired gait, and reduced balance, increasing their risk for injurious falls and fractures [34]. The weight loss seen in frailty could be related to poor nutrition. In a study exploring the correlation of nutrition with frailty, the upper arm circumference, calf circumference, haemoglobin, albumin, prealbumin, cholesterol, and low-density lipoprotein were lower in the frail compared with the non-frail group (*p*  <  0.05) [35]. The risk of frailty was 3.38 times higher in the malnourished compared with the well-nourished patients (*p*  =  0.036). This suggests that nutrition is related to frailty, and malnutrition significantly aggravates the development of frailty. This is likely because of insufficient protein and energy intake, which leads to weight loss and reduced physical function [36]. Malnutrition is also strongly associated with sarcopenia, a condition closely linked to frailty, which leads to reduced physical function and dependency [37]. Another factor linking underweight with frailty is anorexia of ageing. Anorexia of ageing is common and affects up to 30% of older persons, but it could be as high as 85% in residents of care homes [38]. It may be related to impaired senses of taste and smell, which may reduce food intake. Age-associated alteration in stress hormones, increased inflammation, mood disorders such as depression and anxiety, and excess catabolism are risk factors for reduced appetite and anorexia [39]. Other factors, such as an age-related decrease in fundal compliance of the stomach, polypharmacy, and delay in gastric emptying, may result in early and prolonged satiety [40]. In addition, poor oral health, ill-fitting dentures, and being edentulous may also affect chewing ability and food intake, leading to anorexia. Furthermore, social isolation is another important factor associated with decreased appetite and energy intake, which may contribute to the risk of anorexia of ageing [41]. As a result, anorexia will lead to subsequent weight loss, reduced physical activity, and reduction in muscle mass and strength, which eventually increases the risk of the development of frailty [42].

### 3.3. The U-Shaped Relationship

Both low and high body weights are equally associated with frailty, and this may suggest a U-shaped relationship. Data from the English Longitudinal Study of Ageing, which included 3055 community-dwelling older people ≥65 years old, showed a U-shaped relationship between BMI and frailty, with the lowest frailty level in the BMI range between 25 and 29.9 [43]. In another study, to investigate the relationship between frailty and BMI in 7191 older Japanese persons aged ≥65 years, a U-shaped relationship was found, and the BMI range corresponding to the lowest prevalence of frailty was between 21.4 and 25.7 [44]. More recently, a longitudinal study of 1648 non-institutionalised Mexican Americans aged ≥67 years from the Hispanic Established Population for the Epidemiologic Study of the Elderly found that underweight or obese participants had greater odds of frailty over time compared to those with normal weight (OR 2.39, 95% CI 1.29 to 4.44 and 1.62, 1.07 to 2.44, respectively) [45]. Findings from the Epidemiology of Osteoporosis (EPIDOS) Study showed a J-shaped relationship between frailty and BMI. The percentage of underweight and obese women who were frail was 10.8% and 20.3%, respectively [46]. A U-shaped relationship has also been shown in a cohort of 4019 participants aged 41–81 years. The prevalence of physical frailty was 8.2% in underweight participants, 2.9% in normal-weight participants, 2.6% in overweight participants, and 5% in obese participants [31]. In a meta-analysis to explore the relationship between BMI and frailty in community-dwelling older persons aged ≥60 years, the pooled relative risk (RR) of frailty ranged from 1.45 (95% CI 1.10 to 1.90, *p* < 0.01) for the underweight group to 0.93 (0.85 to 1.02, *p* = 0.114) for the overweight group and to 1.40 (1.17 to 1.67, *p* < 0.01) for the obese group compared with normal BMI as the reference group, suggesting a U-shaped relationship [47]. Weight-related factors associated with frailty could be the loss of muscle mass in both underweight and obese individuals, which may lead to sarcopenia and, eventually, the development of frailty. Another factor is the increased inflammation associated with obesity or the catabolic state in persons with weight loss. The association of obesity-related morbid conditions such as cardiovascular disease could be another factor [48].

### 3.4. Frailty and Muscle Mass

Skeletal muscle represents up to 50% of total body weight [49]. With increasing age, the average lifespan decline in muscle mass is 0.37% and 0.47% per year in women and in men, respectively [50]. Frailty may accelerate muscle loss and is associated with lower muscle density and muscle mass but higher fat mass. Data from Invecchiare in Chianti study, which included 923 subjects, mean (SD) age 74.8 (6.8) years, showed that frail subjects had significantly lower muscle density (69.8 mg/cm^3^, SE = 0.4) and muscle area (68.7%, SE = 1.1) compared with non-frail subjects (71.1 mg/cm^3^, SE = 0.2 and 71.2%, SE = 0.4, respectively). Fat area was significantly higher in frail (22%, SE = 0.9) compared with non-frail subjects (20.3%, SE = 0.4) [51]. In a Chinese cross-sectional study of 656 older persons, mean (SD) age of 83.29 (8.29) years, frail subjects had higher waist circumference, body fat mass, and percent body fat compared with non-frail subjects. Skeletal muscle mass (OR 0.159, 95% CI 0.064 to 0.396, *p* < 0.001) was a protective factor for frailty [52]. Reduced muscle mass reduces physical activity, which may result in dysregulation of energy balance, leading to undernutrition that further exacerbates muscle mass loss. The loss of muscle mass, strength, and quality, collectively known as sarcopenia, is closely linked to or a precursor to frailty [53]. Although frailty is a multisystem syndrome, sarcopenia affects only the muscles. Therefore, sarcopenia is considered a part of frailty or, in other words, the muscular manifestation of the frailty syndrome. Sarcopenia can be associated with both malnutrition and weight loss or with weight gain and obesity, a condition called sarcopenic obesity [54]. With increasing age, muscle mass loss is usually paralleled with fat gain, which may accelerate the progression to sarcopenic obesity and frailty [54]. Age-related changes in muscle quality, reduced mitochondrial mass, and increased fat infiltration may contribute to the loss of muscle strength and power [55]. The factors implicated in the development of sarcopenia overlap with those for frailty. For example, a decrease in muscle protein synthesis is a central feature for both conditions. This leads to reduced mitochondrial function and impaired hormonal balance and muscle repair [56].

### 3.5. Frailty and Muscle Fibres

Skeletal muscle mass is responsible for nearly 80% of the total blood glucose uptake [57]. Skeletal muscle tissue consists of different types of muscle fibres that have different contractile and metabolic characteristics. Type I, or the slow twitch, and type II, or the fast twitch, fibres are the most clinically important types that play a role in glucose metabolism. Type II is less oxidative and more glycolytic compared with type I fibres [58]. The low oxidative capacity of type II fibres leads to lower fat oxidation and higher lipid storage in muscle tissue, increasing insulin resistance and predisposing to glucose intolerance and type 2 diabetes. Therefore, type II fibres promote insulin resistance compared with type I fibres. This may provide a potential link between the type of muscle fibres, obesity, and insulin resistance [59]. In other words, the predominance of either type of muscle fibres may influence the overall level of insulin resistance of the individual. Frailty increases the age-related decline in muscle mass, especially type II muscle fibres. Type II muscle fibres are essential for balance maintenance and fall prevention, and the quadriceps muscle strength correlates positively with type II muscle fibre size [60]. Extensive type II muscle fibre atrophy has been shown in frail older women with falls-related hip fractures compared with healthy age-matched controls. Type II muscle fibres were around 30% smaller in women with a hip fracture when compared with healthy age-matched controls (*p* = 0.03) [61]. In a cross-sectional study, frail women displayed a severe reduction in the ratio of type II/I fibre size. Frailty is associated with persistent muscle denervation and neurogenic atrophy. Vastus lateralis muscle biopsies from frail elderly women, mean (SD) age 77.9 (6.2) years, showed significant neurogenic atrophy, whereas that from high-functioning octogenarian female masters athletes, mean (SD) age 80.9 (6.6) years, exhibited superior reinnervation capacity [62]. Sedentary and pre-frail men exhibited complex changes in their vastus lateralis muscle biopsy preferentially affecting type II fibres with a 47% reduction in type II fibre size compared to only a 17% reduction in type I fibre size, as shown in another study [63]. Therefore, it appears that frailty is associated with a more pronounced loss of type II fibres than what could be attributed to age alone, and this may have an impact on insulin resistance and glucose metabolism in frail older persons with diabetes (Table 1).

### 3.6. Frailty and Insulin Resistance

Many studies showed that insulin resistance (IR) and metabolic syndrome are associated with an increased risk of frailty, likely due to an increased risk of cardiovascular complications [64,65,66]. The non-insulin-based IR indicators, such as triglyceride-glucose index, metabolic score for insulin resistance, and estimated glucose disposal rate, were independently associated with the frailty progression and physical function decline [67]. The mechanism could be due to IR-associated dyslipidaemia, chronic low-grade inflammation, oxidative stress, and impaired endothelial function, which contribute to the development of atherosclerosis and cardiovascular disease [68]. However, the reverse direction of the relationship, whether frailty increases IR, is not very clear. There is little literature on the relationship between frailty and IR. Insulin resistance could be the net result of different interacting factors such as visceral obesity, muscle mass, body weight, chronic inflammation, oxidative stress, anti-oxidant capacity, and hormonal status. In a cross-sectional study comparing frail to non-frail subjects, increased abdominal obesity was associated with IR in frail older persons. Non-obese frail subjects have similar IR compared to non-frail subjects. In addition, only abdominal fat mass and muscle mass index were significant predictors, explaining 18.5% and 8.5%, respectively, of the variance in IR. This suggests that body composition, rather than frailty per se, is the determinant factor of IR in frail older people [69]. It has been shown that glucose challenge leads to exaggerated and prolonged insulin responses in frail compared with non-frail subjects, suggesting dysregulation of glucose–insulin dynamics as part of the physiologic vulnerability and reduced reserve associated with frailty [70]. However, frail subjects in this study were significantly more obese than the non-frail group (BMI 28.4 vs. 24.5, *p* = 0.01), which may affect the level of insulin resistance according to the variation in body weight. Animal models showed high IR to be associated with longer survival, suggesting that frailty may be associated with decreased rather than increased IR [71]. Low IR in frail older people, mean (SD) age 85 (7) years, predicted mortality [72]. Low insulin levels have been shown to be associated with sarcopenia and functional measures of sarcopenia, such as body weight, muscle mass, and muscle strength, suggesting that insulin resistance is involved in the pathogenesis of sarcopenia. Low insulin levels may reduce the anabolic effect of insulin on skeletal muscles, leading to sarcopenia [73] (Figure 1).

### 3.7. Frailty Metabolic Spectrum

The relationship between frailty and body weight appears to span from obesity at one end to underweight at the other. In addition, frailty is associated with diverse and complex body composition changes that lead to a metabolic spectrum of variable insulin resistance. In frail individuals, the overall insulin resistance depends on several factors, such as body weight (either gain or loss), the ratio of the insulin-resistant type II to the insulin-sensitive type I skeletal muscle fibres, muscle mass, and total body fat mass, especially the visceral fat. Therefore, the frailty spectrum will start with a sarcopenic obese (SO) phenotype at one end, characterised by increased IR, to an anorexic malnourished (AM) phenotype at the other end with reduced IR. In addition to increased IR due to obesity, the SO phenotype is associated with unfavourable metabolic profiles such as dyslipidaemia, persistent hyperglycaemia, and hypertension, which leads to the progression of metabolic syndrome and acceleration of the progressive course of diabetes [74,75]. On the other hand, due to reduced IR because of significant weight loss, the AM phenotype is associated with reduced prevalence of CV risk factors, reduction in hyperglycaemia, which leads to regression of the metabolic syndrome, or reverse metabolism and deceleration or regression of the course of diabetes. Reverse metabolism is a term used to describe the reverse association of traditional cardiovascular risk factors such as high blood pressure, hyperglycaemia, and hypercholesterolaemia with better survival outcomes compared with low levels. Several studies have indirectly shown the presence of two frailty phenotypes with frail patients who have high HbA1c to be more overweight, obese, have higher waist circumference, higher systolic blood pressure, dyslipidaemia, more use of hypoglycaemic and cholesterol-lowering agents than non-frail cohort [76]. Other studies demonstrated that frail people with low HbA1c have low body weight, low blood pressure, low cholesterol, low serum albumin, low Hb, and lower use of hypoglycaemic and cholesterol-lowering agents than non-frail counterparts [76]. Due to the significant weight loss in the AM phenotype, reduction in hyperglycaemia, normalisation of HbA1c, and successful withdrawal of hypoglycaemic medications can occur, a condition termed frailty-induced burnt-out diabetes [77,78,79]. These patients had significant weight loss, low BMI, low HbA1c, recurrent hypoglycaemia, prevalent dementia, and care home residency. The link between dementia, malnutrition, and frailty has been previously reported, suggesting an increased risk in the AM phenotype, which could set a vicious circle of continuous deterioration [80,81,82]. In between the two polar ends of the frailty spectrum, there will be frail people with a mixed phenotypic and metabolic picture. Some of these frail people will be closer to the AM phenotype, such as having lower muscle mass and lower albumin, but have similar lipid profile, HbA1c, and BMI [83,84]. Other people in the spectrum will be closer to the SO phenotype with high HbA1c, fasting blood glucose levels, and frequent hyperglycaemia but have similar body weight compared with non-frail individuals [85,86].

## 4. SGLT-2 Inhibitors and GLP-1RA

Traditionally, the principal therapeutic and metabolic roles of oral hypoglycaemic therapy were either to increase central insulin secretion (secretagogues) such as sulfonylureas and meglitinides or to increase peripheral insulin utilisation (sensitisers) such as metformin and thiazolidines. The newly introduced agents of SGLT-2 inhibitors and GLP-1RA have novel mechanisms of action that impact body composition, body weight, and insulin resistance. They also have extra-glycaemic prognostic benefits beyond glycaemic control.

### 4.1. Effect on Body Composition

GLP-1RA is associated with improvement in IR in insulin-responsive tissues, especially in obese individuals [87]. This is likely due to the GLP-1RA-induced reduction in chronic inflammation and oxidative stress that improve β-cell function [88,89,90,91]. In a study that included individuals with obesity and prediabetes, the GLP-1RA liraglutide improved insulin sensitivity measured by HOMA–IR for insulin resistance (HOMA-IR) (Homeostatic Model Assessment for Insulin Resistance) as early as 2 weeks prior to weight loss as well as decreasing insulin, C-peptide, and fasting glucagon levels [92]. In another study, dual glucose-dependent insulinotropic polypeptide (GIP) and GLP-1RA, tirzepatide improved markers of insulin sensitivity and beta-cell function to a greater extent than dulaglutide. Improvement in insulin sensitivity was only partly attributable to weight loss, suggesting that the dual receptor agonism confers distinct mechanisms of glycaemic control [93]. In a meta-analysis, GLP-1 RAs were shown to effectively ameliorate IR in patients with non-alcoholic fatty liver disease (NAFLD) [94]. Another meta-analysis, GLP-1RA, showed a significant reduction in body weight, fat mass, and lean mass. The lean mass loss was about 25% of the total weight loss, but the relative lean mass, defined as the percentage change from baseline, was unaffected. Liraglutide was the only GLP-1RA to achieve significant weight loss without significantly reducing lean mass. Tirzepatide and semaglutide were the most effective for weight and fat mass reduction but were among the least effective in preserving lean mass [95].

In a retrospective study to investigate the effects of SGLT-2 inhibitors on body composition, patients in the SGLT-2 inhibitors group showed a greater decrease in body weight, extracellular water, BMI, and waist circumference than patients in the non-SGLT-2 inhibitors group. The bioelectrical impedance analysis revealed a significant decrease in body fat mass and visceral fat area without a loss of lean body mass or skeletal muscle mass in the SGLT-2 inhibitors group. The study concluded that weight loss induced by SGLT-2 inhibitors does not include loss of muscle mass [96]. A meta-analysis showed the beneficial effects of SGLT-2 inhibitors on body composition. SGLT-2 inhibitors reduced body weight, BMI, visceral fat area, subcutaneous fat area, waist circumference, fat mass, and percentage of body fat but also reduced muscle mass. The SGLT-2 inhibitors-induced loss of muscle mass may suggest the inconsistency of the studies on the real effect of SGLT-2 inhibitors on muscle mass. This could be related to variations in study design, different SGLT-2 inhibitors used, duration of the studies, and patients’ characteristics. There is, however, no data on whether SGLT-2 inhibitors can cause sarcopenia [97]. By reducing hyperglycaemia, SGLT-2 inhibitors mitigate the effect of glucotoxicity on the β-cells and reduce IR [98]. A cross-sectional study to investigate the effects of SGLT-2 inhibitors on insulin secretion and resistance showed improvement in β-cell function and a 21% (*p* = 0.008) reduction in insulin resistance [99]. The improvement in β-cell function and IR have also been shown in other studies [100,101].

### 4.2. Current Limitations of Clinical Guidelines

The benefits of SGLT-2 inhibitors and GLP-1RA extend to include older persons. About 50% of participants in clinical trials of these agents were ≥65 years of age. The post hoc analyses of the EMPA-REG OUTCOME study (44.6% of participants ≥65 years) and of the DECLARE study (46% of participants ≥65 years) have found that the risks of cardiovascular mortality, heart failure, and renal outcomes were reduced across all age groups [102,103]. In the age-stratified meta-analysis of the SGLT-2 inhibitors clinical trials, the therapeutic benefit was consistent across all age groups [104]. Therapeutic efficacy also extended to include older persons ≥75 years, as demonstrated in the post hoc analysis of the LEADER study [105]. These findings are limited by the small number of participants above the age of 75 years, the exploratory nature of the post hoc analyses, and the fact that frailty was not tested in these trials. In addition to efficacy in older age groups, these agents appear to be well-tolerated with a mild increase in side effects compared to younger people [105]. These agents showed a low risk of hypoglycaemia, similar to placebo arms, which makes them appealing to frail older persons with diabetes [106]. The guidelines generally recommend relaxed glycaemic targets in older persons with multiple morbidities [7,8,107,108,109,110,111,112]. Frailty is not well defined in clinical guidelines and is always referred to as one category. Although the physical phenotype or the accumulation of deficits tools are usually used to diagnose frailty, the guidelines do not recognise the need for identifying the metabolic profile of frailty or its use in choosing the appropriate hypoglycaemic therapy. The effect of frailty on body composition and its impact on the metabolic profile of the individual is also less well-recognised. The metabolic heterogeneity of frailty is not described, and the recommendation for the use of SGLT-2 inhibitors or GLP-1RA does not take into consideration the metabolic phenotype of frailty. The focus of the guidelines is to avoid inducing side effects, such as weight loss and dehydration, in frail people, and no clear guidance on the benefits of such therapy in obese frail older persons. In addition, glycaemic targets and goals of therapy in different metabolic categories of frail persons are not clearly specified (Table 2).

### 4.3. Use in Frailty

Although about 50% of the participants in the clinical trials of SGLT-2 inhibitors and GLP-1RA were ≥65 years old, these trials did not assess for or include people with frailty. Therefore, there is no direct evidence of efficacy and safety of these agents in frail older persons with diabetes. These new drugs, although initially developed as hypoglycaemic agents, are more metabolically active, improve metabolic profile lead to positive outcomes independent of glycaemic control. Therefore, frail, obese older persons with diabetes are likely to benefit most from these agents compared to anorexic malnourished individuals who are more likely to develop side effects such as further weight loss, dehydration, and hypotension. There is recently published indirect evidence suggesting that SGLT-2 inhibitors and GLP-1RA are effective in frail compared to non-frail older persons with diabetes, and the benefits increase with the increase in severity of frailty [113]. Data from the retrospective analysis of 744,310 older patients ≥65 years of age with type 2 diabetes mellitus showed an absolute CV risk reduction among frail people. In patients treated with SGLT-2 inhibitors, incidence rate difference (IRD) was −27.24 (95% CI −41.64 to −12.84) in frail and −6.74 (−8.61 to −4.87) in non-frail compared with dipeptidyl peptidase-4 (DPP-4) inhibitors. Similarly, GLP-1RA therapy showed an IRD of −25.88 (−38.30 to −13.46) in frail and −7.02 (−9.23 to −4.81) in non-frail compared with DPP-4 inhibitors. This high absolute risk reduction translates into an impressive number needed to treat (NNT) to prevent CV events over 1 year of 39 vs. 159 for SGLT-2 inhibitors and 42 vs. 162 for GLP-1RA compared with DPP-4 inhibitors [114]. Most of the frail compared to non-frail patients included in this study were overweight or obese (61.68% vs. 39.01% for the SGLT-2 inhibitors cohort) and 66.54% vs. 44.60% for the GLP-1RA cohort). In addition, frail patients had unfavourable metabolic profiles such as an increased prevalence of dyslipidaemia, hypertension, chronic kidney disease, and fatty liver. Moreover, they have a more prevalent history of CV events compared to non-frail cohorts. This explains the high absolute risk reduction in frail subjects, reflecting their high baseline risk and their unfavourable metabolic status due to obesity and high IR [114]. On the other hand, this study excluded patients living in care homes or those who had had an episode of skilled nursing home admission in the previous 365 days prior to the study entry. Nursing home residents are likely to be predominantly AM frail phenotype, with significant weight loss due to the fact that institutionalisation itself increases the risk of anorexia and malnutrition [115].

### 4.4. Clinical Implications

It appears from the limited available evidence that frail older persons on the obese side of the frailty spectrum will benefit from SGLT-2 inhibitors and GLP-1RA therapies. The benefit is even higher in individuals with severe frailty, at the obese end of the frailty spectrum, due to the unfavourable metabolic profile, progression of the metabolic syndrome, and increased IR.

Therefore, the sarcopenic obese frail individuals are likely to have high atherosclerotic cardiovascular disease (ASCVD) risk and qualify for these therapies as a first line, in combination with metformin [116]. The addition of such therapy to metformin should not be driven by HbA1c, as the benefits of these agents are independent of glycaemic control. However, because of sarcopenia in these obese frail individuals and the potential negative effects of these agents on muscle mass, education about the importance of resistance exercise training and adequate nutrition will be required at the start of therapy. Progressive resistance training exercise combined with a protein-rich diet can result in an increase in muscle mass, strength, and muscle performance [117]. On the other hand, there is no available evidence in the literature to confirm the benefits of these agents in the anorexic malnourished side of the frailty spectrum. For example, frail patients living in nursing homes were either excluded or minimally presented in clinical trials. In addition, patients in whom these new therapies were discontinued, compared to those who tolerated them, were significantly older, mean (SD) age of 75.8 (4.2) vs. 74.7 (3.8) years, *p* < 0.002 and had a lower BMI of 27.9 (3.3) vs. 29.2 (4.7). Moreover, discontinuations due to intolerance were two times higher in the very old (≥80 years) compared to the younger patients (19.1% vs. 35%, *p* = 0.005), and lower BMI significantly predicted drug intolerance (OR 0.92, 95% CI 0.88 to 0.97, *p* < 0.001) [118]. In addition to medication choice, across the frailty spectrum, the goals of therapy should be individually tailored. For example, due to the high ASCVD burden in the obese end of the spectrum, the focus will be on intensification of therapy, and the goal is CV risk reduction. On the anorexic side of the spectrum, the focus will be on the deintensification of therapy to reduce the risk of hypoglycaemia, and the goal is to control the symptoms and maintain the quality of life (Figure 2).

## 5. Conclusions

Frailty is an increasingly recognised complication of diabetes in older persons. Frailty is associated with diverse body composition changes that affect the overall metabolic profile and insulin resistance of the individual. Following the introduction of SGLT-2 inhibitors and GLP-1RA with novel mechanisms of action, the beneficial outcomes depend largely on the metabolic effects induced by these new agents rather than their effect on glycaemic control. Therefore, the metabolic profile of the frail individual should be considered when prescribing these new agents to help clinicians practice personalised and precision medicine.

## 6. Future Perspectives

The relationship between frailty and body composition is complex. The association of frailty with various body composition changes leads to a metabolic spectrum of frailty that exhibits variable insulin resistance, depending on the relative muscle mass, predominant muscle fibre type, and visceral fat ratios, in addition to the total body weight. This spectrum ranges from individuals with significant weight loss due to anorexia and malnutrition leading to reduction in insulin resistance to obese individuals with increased visceral fat leading to increased insulin resistance. These changes will affect the choice of the new glucose-lowering therapy, which is metabolically active rather than traditionally hypoglycaemic agents. Confirmation of the metabolic heterogeneity of frailty is still required in future research. It is likely that obese frail individuals will benefit most from these agents therefore this needs future investigation.

Future studies should consider the demographic and metabolic subtypes of frail participants rather than referring to frailty as one homogeneous group. A randomised clinical trial of SGLT-2 inhibitors and GLP-1RA in frail older persons stratified by metabolic subtypes is required. In addition to exploring these new agents as monotherapy in frail participants, there is evidence to suggest that the effects of SGLT-2 inhibitors and GLP-1RA are complimentary with the superiority of SGLT-2 inhibitors in reducing the incidence and hospitalisation of heart failure, preserving renal function and reducing body weight. On the other hand, GLP-1RA is superior in reducing the risk of stroke and improving unfavourable metabolisms, such as glucose dysregulation and dyslipidaemia [119,120]. Therefore, the use of combination therapy, in addition to metformin, in obese frail individuals who are likely to have a high burden of ASCVD risk should be explored [121,122,123]. Initiation of a triple therapy concept consisting of metformin, SGLT-2 inhibitors, and GLP-1RA, which will have superior positive outcomes than each agent alone, needs to be tested in these high-cardiovascular-risk patients. Some frail patients on the low-weight side of the frailty spectrum may benefit from such therapy. Future studies need to investigate whether the addition of insulin to the new therapy in these patients will avoid further weight loss because of the weight-gaining properties of insulin. Due to its anabolic properties, insulin therapy may improve skeletal muscle parameters and mitigate any negative effects of the new agents on muscle mass or function [124]. Finally, there is a reluctance by health care professionals to use these new agents in a care home setting [125]. This is likely due to the fact that many patients in these settings are malnourished with weight loss and are not suitable for such therapy. However, obese frail individuals living in care homes may benefit from these agents. The occurrence of positive outcomes of these agents in a short time period makes them suitable for these patients, although they may have short life expectancy. Therefore, increasing health care professionals’ awareness of which patients in these settings are suitable for such therapy is required.

## 7. Key Points

Frailty has a diverse and complex association with body composition.Frailty-induced body composition changes lead to the emergence of a metabolic spectrum of frail individuals with variable insulin resistance.The insulin resistance of frail patients will depend on the net effect of visceral fat and muscle mass ratio, dominant skeletal muscle fibre type, and total body weight.SGLT-2 inhibitors and GLP-1RA are metabolically active rather than hypoglycaemic agents; therefore, their effects will be different across the frailty spectrum.The obese frail are likely to benefit most from these agents due to the prevalence of high insulin resistance, while anorexic frail are likely to be intolerant to such therapy.

## Figures and Tables

**Figure 1 metabolites-15-00381-f001:**
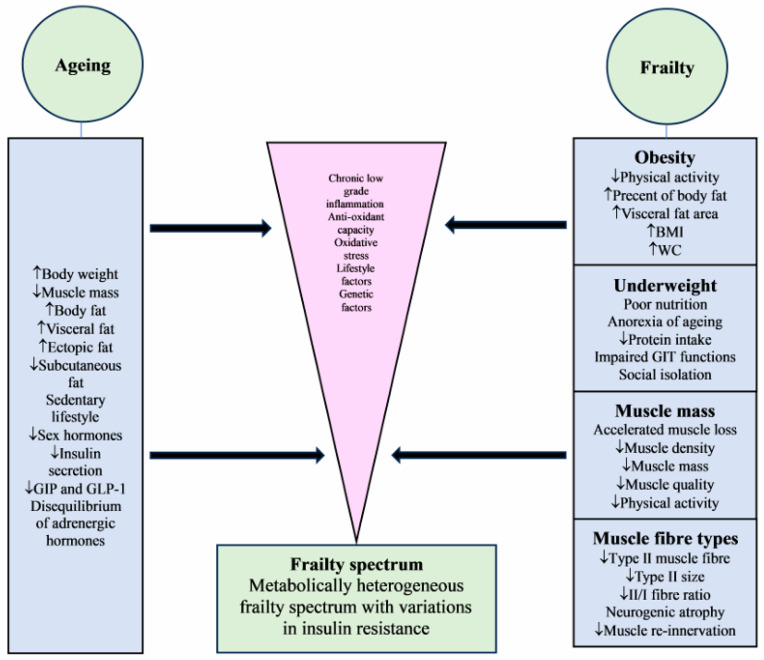
Effects of ageing and frailty on body composition leading to a heterogenous metabolic spectrum of frailty. GIP = gastric inhibitory peptide, GLP-1 = glucagon-like peptide-1, BMI = body mass index, WC = waist circumference, GIT = gastrointestinal tract.

**Figure 2 metabolites-15-00381-f002:**
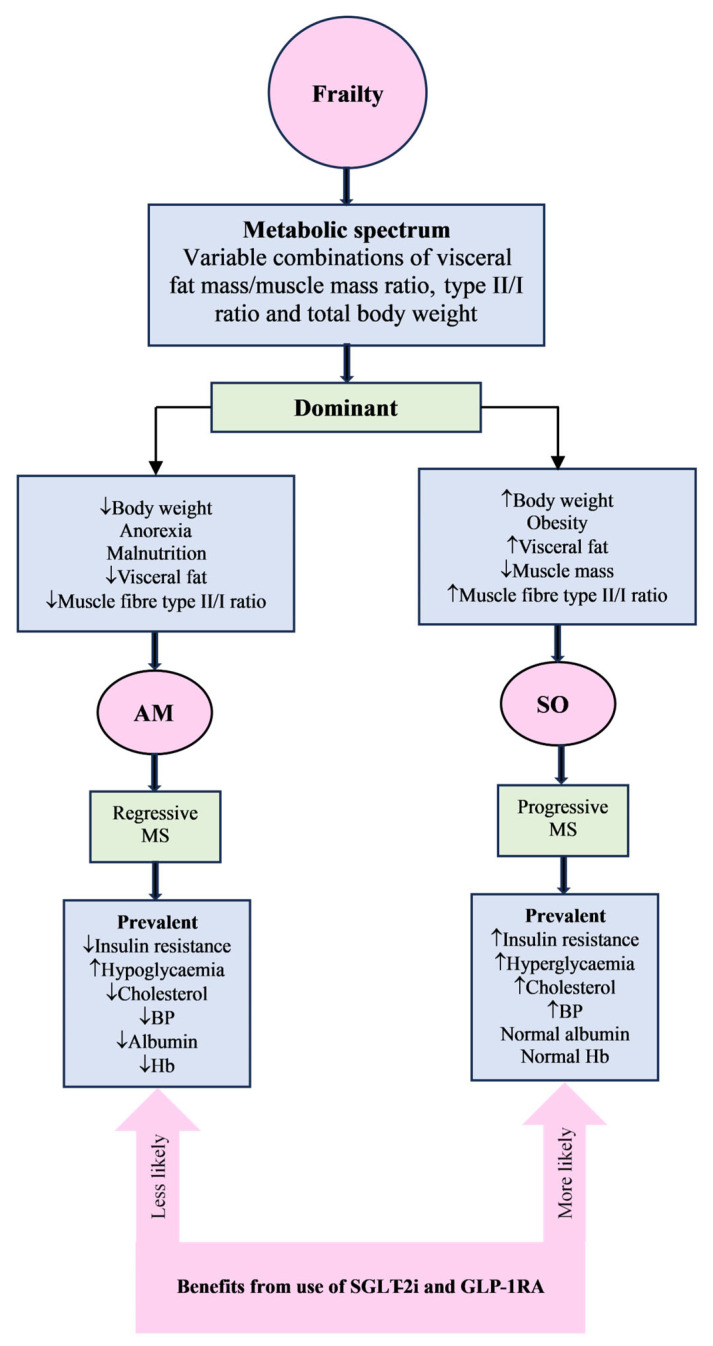
The metabolic spectrum of frailty, the metabolic profile of the two polar phenotypes, and the likely benefits of SGLT-2i and GLP-1RA. AM = anorexic malnourished, SO = sarcopenic obese, MS = metabolic syndrome, BP = blood pressure, Hb = haemoglobin, SGLT-2i = sodium glucose cotransporter-2 inhibitors, GLP-1RA = glucagon-like peptide-1 receptor agonists.

**Table 1 metabolites-15-00381-t001:** Metabolic characteristics of the main skeletal muscle fibre types *.

Type I	Type II
Red in colour, slow twitching and fatigues slowly.	Pale in colour, fast twitching and fatigues rapidly.
Has high aerobic oxidative capacity.	Has low anaerobic oxidative capacity.
Has high blood supply, high capillary and mitochondrial number.	Has low blood supply, low capillary and mitochondrial number.
Low lipid content.	High lipid content.
Low glycolytic capacity.	High glycolytic capacity.
Low insulin resistance.	High insulin resistance.
Declines less with increasing age and frailty.	Declines more with increasing age and frailty.

* The relative dominance of skeletal muscle fibre type in frail older persons with diabetes contributes to the overall individual insulin resistance.

**Table 2 metabolites-15-00381-t002:** Guidelines’ inclusion of frailty and recommendations of SGLT-2i and GLP-1RA in frail older persons with type 2 diabetes mellitus.

Guidelines	Frailty Inclusion	Recommendations
ADA [7]	Frailty is not defined, but screening for frailty, as a part of geriatric syndromes, is recommended.	SGLT-2i: Caution in individuals depending on caregivers for adequate fluid intake or have recurrent UTI. GLP-1RA: Injectables, which require visual, motor, and cognitive skills for administration, may not be preferred in people experiencing unexplained weight loss, suspected gastroparesis, or recurrent gastrointestinal problems.
ESE [8]	Frailty is defined, and screening using frailty screening tools is recommended.	SGLT-2i: Limit the dose in patients at risk of volume loss. GLP-1RA: Nausea is a common side effect and could be problematic in patients with compromised intake, especially those with progressing CKD.
IDF [107]	Defined and screening by frailty tool is recommended.	SGLT-2i: Associated with increased risk of genital and urinary tract infections, hypovolaemia, postural hypotension, and weight loss may limit their use in some older people. GLP-1RA: May not be appropriate for frail older people in whom weight loss can be detrimental.
IPS [108]	Frailty is defined, and screening using various frailty tools is recommended.	SGLT-2i: Watch for increased urinary frequency, incontinence, lower BP, genital infections, dehydration, and dose reduction required in the presence of renal impairment. GLP-1RA: Monitor for anorexia, weight loss, and dose reduction needed in moderate impairment.
EuGMS/EDWPOP [109]	Frailty is defined, and screening by easy-to-perform and validated tool, which does not necessarily require professional staff, is recommended as integral part of routine care.	SGLT2i: Not suitable for moderate–severe frailty or care home residents with weight loss. They increase risk of UTI, candidiasis, dehydration, hypotension, and diabetic ketoacidosis. GLP-1RA: Not suitable for subjects with CKD or care home residents with weight loss.
Diabetes Canada [110]	Defined and screening by frailty tool is recommended.	SGLT-2i: Could be considered for people < 75 years with evidence of CVD, relatively preserved renal function, and no other complex comorbidities. GLP-1RA: Older people may be more susceptible to dehydration and fractures than younger people treated with these agents, suggesting that they should be used cautiously.
JDS [111]	Not defined but recommend screening as part of geriatric syndromes.	General recommendation: SGLT-2i: Attention to dehydration and urogenital infections. GLP-1RA: Attention to gastrointestinal symptoms (nausea and vomiting) and weight loss.
KDS [112]	Defined and screening by frailty tools is recommended.	General recommendation: SGLT-2i: Caution due to risk of dehydration and weight loss. GLP-1RA: Their side effects significantly increase in people over 60 years without ASCVD.

SGLT-2i = sodium glucose cotransporter inhibitors, GLP-1RA = glucagon-like receptor agonists, ADA = American Diabetes Association, UTI = urinary tract infection, ESE = European Society of Endocrinology, CKD = chronic kidney disease, IDF = International Diabetes Federation, IPS = international position statement, EuGMS = European Geriatric Medicine Society, EDWPOP = European Diabetes Working Party for Older People, CVD = cardiovascular disease, JDS = Japanese Diabetes Society, KDS = Korean Diabetes Society, ASCVD = atherosclerotic cardiovascular disease.

## Data Availability

No new data were created or analysed in this study.

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
