# Peer review of "The Effect of Frailty on Body Composition and Its Impact on the Use of SGLT-2 Inhibitors and GLP-1RA in Older Persons with Diabetes"

_metabolites, 2025, doi:10.3390/metabo15060381_

Round 1
Reviewer 1 Report
Comments and Suggestions for Authors
The manuscript is very well-structured and presents a compelling investigation into a significant area. The importance of this topic is evident, and the authors have done an excellent job in their presentation.I have two minor points for the authors to consider:
Terminology: For consistency and precision in scientific writing, I recommend using "persons with diabetes" rather than "people with diabetes" throughout the text.
Graphical Features: While the information conveyed in the figures is clear, some minor adjustments to their design (e.g., font sizes, color choices) could further improve their readability and impact.
Author Response
The manuscript is very well-structured and presents a compelling investigation into a significant area. The importance of this topic is evident, and the authors have done an excellent job in their presentation. I have two minor points for the authors to consider:
Terminology: For consistency and precision in scientific writing, I recommend using "persons with diabetes" rather than "people with diabetes" throughout the text.
Graphical Features: While the information conveyed in the figures is clear, some minor adjustments to their design (e.g., font sizes, color choices) could further improve their readability and impact.
Many thanks for your comments and suggestions to improve the manuscript.
- We have changed to persons with diabetes throughout the manuscript.
- We reviewed the figures as suggested. The figures appear better in our word document but, for some reason, it does not appear complete in the version sent by the journal. We will contact the journal for this point and, as we have no specific preference to colour choice, we will leave this option to the journal to decide on best colours suits.
Reviewer 2 Report
Comments and Suggestions for Authors
Dear Author,
The review "The effect of frailty on body composition and its impact on the use of SGLT-2 inhibitors and GLP-1RA in older people with diabetes" is interesting however the overall presentation of the review if you can please address the following comments.
- Explicitly state the central thesis in the Introduction (e.g., “Frailty-associated metabolic heterogeneity dictates SGLT-2i/GLP-1RA suitability in older adults”).
- Combine “Body Composition Aging Effect” and “Frailty Effect” to eliminate overlap in muscle/fat loss discussions.
- Resolve conflicting claims about SGLT-2i effects on lean mass (Page 8 vs. Page 9 meta-analysis).
- Add a column to Table 2 comparing guideline alignment with the proposed frailty spectrum.
- Improve Figure 2: Label axes (e.g., “Insulin Resistance Level”) and clarify “ICI>olesterol” typo.
- Define “Reverse Metabolism”: Briefly explain this term (Page 8) for readers unfamiliar with the concept.
- In “Future Perspectives,” suggest a concrete study (e.g., “Randomized trial of SGLT-2i/GLP-1RA in sarcopenic obese vs. anorexic frail phenotypes”).
- The similarity index is high please reduce that.
best regards,
Comments on the Quality of English Language
The English could be improved to more clearly express the research.
Author Response
The review "The effect of frailty on body composition and its impact on the use of SGLT-2 inhibitors and GLP-1RA in older people with diabetes" is interesting however the overall presentation of the review if you can please address the following comments.
- Explicitly state the central thesis in the Introduction (e.g., “Frailty-associated metabolic heterogeneity dictates SGLT-2i/GLP-1RA suitability in older adults”).
- Combine “Body Composition Aging Effect” and “Frailty Effect” to eliminate overlap in muscle/fat loss discussions.
- Resolve conflicting claims about SGLT-2i effects on lean mass (Page 8 vs. Page 9 meta-analysis).
- Add a column to Table 2 comparing guideline alignment with the proposed frailty spectrum.
- Improve Figure 2: Label axes (e.g., “Insulin Resistance Level”) and clarify “ICI>olesterol” typo.
- Define “Reverse Metabolism”: Briefly explain this term (Page 8) for readers unfamiliar with the concept.
- In “Future Perspectives,” suggest a concrete study (e.g., “Randomized trial of SGLT-2i/GLP-1RA in sarcopenic obese vs. anorexic frail phenotypes”).
- The similarity index is high please reduce that.
Many thanks for your comments and suggestions to improve the manuscript.
- Frailty-associated metabolic heterogeneity etc, added in the introduction.
- Combine ageing and frailty effect on body composition. We have left ageing effect separate. The frailty effect on body composition is large and a main part of the manuscript and followed by several subheadings. We have made it a bit more clear by linking the heading to subheadings (highlighted).
- Conflict of SGLT-2i effects on lean mass etc, explanation added, page 9.
- We are not very clear what is actually required in this point regarding adding a column to table 2. If it meant for us to suggest new guidance based on the metabolic profile of frailty, it would not be possible to suggest this on behalf of every individual society as each society will have their own recommendation. This clinical recommendation is outside the scope of this manuscript. However, we are happy to do any further change if you think we have misunderstood your point.
- Apology, both figures are not clear in the version sent by the journal. This will be corrected by direct communication with the journal.
- Reverse metabolism, defined, page 8.
- Randomized trial of SGLT-2i/GLP-1RA in sarcopenic obese vs. anorexic frail phenotypes, added in future perspectives, page 14.
- Similarity index, we are aware of that. It is likely related to our previous publication in the same topic. We tried to improve this as possible.